# Magnetic Bead-Based Electrochemical Immunoassays On-Drop and On-Chip for Procalcitonin Determination: Disposable Tools for Clinical Sepsis Diagnosis

**DOI:** 10.3390/bios10060066

**Published:** 2020-06-17

**Authors:** Águeda Molinero-Fernández, María Moreno-Guzmán, Miguel Ángel López, Alberto Escarpa

**Affiliations:** 1Department of Analytical Chemistry, Physical Chemistry and Chemical Engineering, Universidad de Alcalá, Ctra. Madrid-Barcelona, Km. 33.600, Alcalá de Henares, 28871 Madrid, Spain; agueda.molinero@edu.uah.es; 2Department of Chemistry in Pharmaceutical Sciences, Analytical Chemistry, Faculty of Pharmacy, Universidad Complutense de Madrid, Avenida Complutense, s/n, 28040 Madrid, Spain; marimore@ucm.es; 3Chemical Research Institute “Andres M. Del Rio”, Universidad de Alcalá, Alcalá de Henares, 28871 Madrid, Spain

**Keywords:** electrochemical immunoassays, microfluidic chips, screen-printed, sepsis

## Abstract

Procalcitonin (PCT) is a known protein biomarker clinically used for the early stages of sepsis diagnosis and therapy guidance. For its reliable determination, sandwich format magnetic bead-based immunoassays with two different electrochemical detection approaches are described: (i) disposable screen-printed carbon electrodes (SPE-C, on-drop detection); (ii) electro-kinetically driven microfluidic chips with integrated Au electrodes (EMC-Au, on-chip detection). Both approaches exhibited enough sensitivity (limit of detection (LOD) of 0.1 and 0.04 ng mL^−1^ for SPE-C and EMC-Au, respectively; cutoff 0.5 ng mL^−1^), an adequate working range for the clinically relevant concentrations (0.5–1000 and 0.1–20 ng mL^−1^ for SPE-C and EMC-Au, respectively), and good precision (RSD < 9%), using low sample volumes (25 µL) with total assay times less than 20 min. The suitability of both approaches was successfully demonstrated by the analysis of human serum and plasma samples, for which good recoveries were obtained (89–120%). Furthermore, the EMC-Au approach enabled the easy automation of the process, constituting a reliable alternative diagnostic tool for on-site/bed-site clinical analysis.

## 1. Introduction

Nowadays, the burden of sepsis on health care is highly significant. It is estimated that approximately 13 million people worldwide become septic every year and four million people die of sepsis. Mortality rates for severe sepsis range between 30 to 50%, and higher than 50% for septic shock [1]. Furthermore, despite advances in health care, the incidence of sepsis is increasing every year and a continuous increment is expected as the population ages [2]. With this in mind, new diagnostic tests that help clinicians to diagnose and manage this disease can significantly yield improvements in their patients’ outcomes.

Procalcitonin (PCT) is a protein precursor of the calcitonin hormone, composed of 116 amino acids, with a molecular weight of 13 kDa. It is considered a very specific biomarker in early clinical diagnosis for severe infection diseases, including sepsis [3]. Under normal physiological conditions, PCT levels in human blood is lower than 0.25 ng mL^−1^, but it can rapidly increase in response to pro-inflammatory stimulation, with a half-life of 24 h [4,5]. Measurements of PCT levels could be used for diagnosis, as well as for evaluations of the treatment effectiveness, determining the appropriate dosage and duration of antibiotic therapy [6,7,8].

Several methods have been reported for PCT determination, mainly based in the selective recognition provided by immunological interactions. These methodologies include immunoturbidimetric assays [9], chemiluminescent immunoassays [10,11,12,13,14,15,16], immunochromatographic assays [17,18,19,20,21,22], surface plasmon resonance biosensors [23,24,25], fluorescence immunosensors [26,27,28,29,30,31,32], ellipsometry immunosensors [33], colorimetric immunoassays [34,35,36], and electrochemical immunoassays [37,38,39,40,41,42,43,44,45,46,47,48,49,50,51,52,53,54,55,56]. Even different immunoanalytical methodologies are commercially available by Brahams GmbH (Henningsdorf, Germany). Although some of them reach impressive sensitivity, the vast majority present several limitations, such as their high complexity, sophisticated instrumentation requirements and/or unproven applicability in the real clinical scenario. 

Furthermore, among all the detection systems, electrochemistry stands out due to its inherent miniaturization, portability, low cost and ability to tailor electrode materials [57] The coupling of this detection principle has been widely explored in immunoassays for clinical diagnoses. In particular, electrochemical magneto-immunosensors have proven to be sensitive, accurate, fast and inexpensive, capable of achieving adequate limits of detection and very suitable as point-of-care tools for decentralized analysis. The use of magnetic beads has been largely exploited in recent years due to their well-known properties such as improved assay kinetics and easy manipulation, while they are perfectly coupled to electrochemical detection [58]. However, magneto-immunosensors [16,17,35,36] in general, and particularly those using electrochemical detection [40], have not been widely explored for PCT determination.

Keeping in mind the aforementioned characteristics of electrochemistry, it can be considered as a very suitable detection technique to be interfaced with microfluidics as well. In addition, microfluidics enable lower sample consumption, as well as finely controlled and automated (electro)-chemical reactions in pocket-sized devices containing microchannels to provide point-of-care applications [59,60]. In this sense, they constitute an ideal platform to perform integrated microscale immunoassays [59,60,61,62,63,64]. Despite these inherent advantages, few microfluidic immunosensors for PCT determination have been previously published, and most of them are based on optical techniques [26,28,29,30,39].

In this work, we demonstrate the use of magnetic bead-based electrochemical immunoassays for PCT determination in human serum samples using two approaches: (i) disposable screen-printed carbon electrodes (SPE-C, on-drop detection) and (ii) electro-kinetically driven microfluidic chips with integrated Au electrodes (EMC-Au, on-chip detection).

## 2. Materials and Methods

### 2.1. Reagent and Solutions 

The analyte, procalcitonin (PCT) (8PC5), and biotinylated and horseradish peroxidase (HRP)-conjugated monoclonal anti-PCT antibodies(18B7,44D9) were purchased from HyTest (Turku, Finland).

The lyophilized PCT was dissolved in deionized water (Millipore Milli-Q purification system). Further PCT dilutions were carried out with 0.1 M phosphate buffered saline and 0.01% Tween-20 (PBST) buffer, pH 7.5. Antibody solutions were prepared in 0.1 M phosphate buffered saline and 0.01% Tween-20 (PBST) buffer, pH 7.5.

Streptavidin-coated superparamagnetic beads (Dynabeads^®^ M-280 Streptavidin) (10 mg mL^−1^) were obtained from Invitrogen (Carlsbad, CA, USA), and bovine serum albumin (BSA), hydroquinone (HQ) and 30% H_2_O_2_ PERDROGENTM (w/w) were purchased from Sigma-Aldrich (Madrid, Spain). Hydroquinone and hydrogen peroxide solutions for the electrochemical detection were prepared in 0.1 M of phosphate-buffered (PB) solution, pH 7.0. 

### 2.2. Apparatus and Electrodes

Magnetic racks for magnetic bead immobilization onto the working electrode surface were purchased from Metrohm DropSens (Oviedo, Spain). A magnetic block, DynaMag™-2, and a Sample Rack for DynaMag™-2 for magnetic particle handling were purchased from ThermoFisher, (Carlsbad, CA, USA). For the incubation steps, a Vortex Mixer-ZX3 from Velp Scientifica and a Thermoshaker TS-100C from Biosan, (Riga, Latvia) were used.

Multi potentiostat/galvanostat μSTAT 8000 and “DropView 8400” software for measurement setup and data acquisition, handling, processing, and exporting was used for the on-drop amperometric measurements. Screen-printed carbon electrodes (SPCE) DRP-110, with a carbon working electrode (ø = 4 mm), carbon counter electrode and silver reference electrode, from Metrohm DropSens (Oviedo, Spain), were used (Appendix A). A holder (MCE-HOLDER-DC02) and microchips (MCE-SU8-Au002T) (38 × 13 × 0.75 mm) from MicruX Technologies S.L. (Oviedo, Spain) were used for on-chip measurements. The microchips integrate three Au electrodes of 100 µm (working electrode (WE), auxiliary electrode (AE) and counter electrode (CE)) with a separation channel length of 30 mm and an injection channel length of 5 mm. The width of the microchannel is 50 µm and their depth is 20 µm. The microchip was placed into a holder, where all electric contacts and reservoirs were pre-defined (Appendix A).

For microchip pre-treatment, 0.1 M NaOH solution was flushed through the channels for 20 min, followed by rinsing with deionized water for 10 min and PBST 0.1 mM (running buffer) for 10 min.

The Bi-potentiostat HVSTAT2010, for applying the high-voltages and recording the amperometric measurements, was obtained from MicruX Technologies S.L. (Oviedo, Spain). 

### 2.3. Samples

Human samples from healthy volunteers, with undetectable PCT levels, were obtained after their written informed consent and authorization. 

This study was conducted in accordance with the Declaration of Helsinki Ethical Principles, and was approved by the Ethics Committee of the Hospital Clínico San Carlos (Spain) (reference code: C.P.—C.I. 16/161-E. Date of approval: 23 May 2016).

### 2.4. Immunoassay Procedures

Based on a typical sandwich ELISA protocol, 2 µL of the commercial streptavidin-coated magnetic bead (MBs) suspension was placed into a microcentrifuge tube, and subjected to a washing step according to the manufacturer’s protocol. These beads were incubated in 50 µL (5 µg mL^−1^) of biotinylated anti-PCT solution in PBST buffer at room temperature and stirred for 5 min. After that, the microcentrifuge tube was placed on the magnetic block and the supernatant was removed, followed by two washing steps with 100 µL of PBST buffer. Then, the MBs functionalized with anti-PCT antibody were re-suspended in 25 µL of sample or PCT standard solutions, plus 25 µL of HRP-conjugated anti-PCT antibody solution (0.36 µg mL^−1^, final concentration) in PBST with 0.1% BSA. After the suspension incubation at room temperature for 15 min, the supernatant was removed and three washing steps were carried out.

Once the immunoreaction was carried out, electrochemical detection was performed using both approaches: (i) on-drop onto screen-printed carbon electrodes (SPE-C); (ii) on-chip into electro-kinetically driven microfluidic chips with integrated Au electrodes (EMC-Au) (Figure 1).

### 2.5. Electrochemical Detection On-Drop onto SPE-C

After the immunocomplex formation, the MBs were re-suspended in 1-mM (45-μL) hydroquinone solution and transferred to the SPCE, where they were placed onto the working electrode surface using a magnet. Finally, amperometric measurements were performed at an applied potential of −0.20 V. After current stabilization, 5 µL of hydrogen peroxide solution (final concentration = 5 mM) was added, and the current was recorded. 

The amperometric signals were calculated as the difference between the steady-state and the background currents at 200 s (Appendix A). The signals were then fit to a four-parameter logistic regression using SigmaPlot 10.0 (Equation (1)).
(1)$$\mathrm{ip}=\left(\frac{i_{max}-i_{min}}{1+{\left(\frac{EC_{50}}{x}\right)}^{h}}+i_{min}\right)$$
where *i_max_* and *i_min_* are the maximum and minimum current values of the calibration graph; *EC*_50_ value is the analyte concentration corresponding to 50% of the maximum signal; *h* is the hill slope.

### 2.6. Electrochemical Detection into EMC-Au

In this case, the MB immunocomplexes were re-suspended in 10 µL of PBST buffer for their subsequent electro-kinetical introduction into the microfluidic chip. Therefore, this suspension was deposited into the sample reservoir (SR) of the microfluidic chip (Figure 2). In addition, microchannels, the running buffer and detection reservoirs (RB and DR) were filled with PBST, while the enzymatic substrate reservoir (ER) was filled with a mixture of 45 µL of 1 mM HQ plus 5 µL of 50 mM H_2_O_2_.

An electrokinetic injection protocol was optimized for the EMC-Au electrochemical detection (Figure 2). MBs were dragged to the longitudinal channel, applying a voltage of +1500 V between reservoirs SR and DR for three pulses of 25 s, while other reservoirs were left floating. They were retained within the microchannel by the aid of a magnet situated on the top. After a washing step with PBST (10 s applying +1500 V from RB to DR reservoirs) the enzymatic substrates were injected and pumped to cross through the particle bed (200 s applying +1500 V from ES to DR reservoirs). In-channel amperometric measurements were taken at an applied potential of −0.20 V on the Au working electrode. The amperometric signals were calculated as the difference between the steady-state and the background currents at 200 s (Appendix A) and fit to a four-parameter logistic regression (Equation 1) using SigmaPlot 10.0. After the measurement, MBs were removed from the main channel by taking off the magnet and washing the channel by injection of buffer for 200 s (+1500 V) from RB to DR. 

Taking into account that only a small fraction of the MBs deposited into the sample reservoir are introduced into the main channel, the analysis can be automatically repeated several times without the need for manual intervention or conditioning of the microchip.

## 3. Results and Discussion

### 3.1. Optimization of the Immunoassay

The functionalization of the MBs with the biotinylated captured antibody (cAb) was evaluated in a concentration of antibodies between zero and 7.5 µg mL^−1^. The amount of cAb depends on the number of MBs used and the number of streptavidin molecules immobilized onto them. The maximum current intensity was obtained using a concentration of 5.0 µg mL^−1^, followed by a plateau that denotes the saturation of the binding sites (Appendix A). A similar selection protocol was followed for the determination of the optimal concentration of the detection antibody. Titration was performed for concentrations ranging from 0.04 to 0.7 µg mL^−1^, where the maximum intensity current was reached for 0.36 µg mL^−1^ of anti-PCT-HRP producing the saturation of the antigen/capture-antibody binding sites (sandwich format) (Appendix A). Incubation times were also studied for different stages. Times of 5 min for the immobilization of captured antibodies to modified magnetic beads produced 85% of the maximum intensity current (Appendix A). Moreover, the simultaneous or sequential incubation of the analyte and detection antibody was also considered. In total, 97% of the maximum current was obtained when simultaneous incubation of both species was performed for 15 min (Appendix A). Non-specific adsorption was almost negligible (<1%) when adding 0.1% BSA to the dilution buffer during the incubation stages.

Once the immunorecognition was performed, the electrochemical detection was carefully studied using two different approaches: (i) SPE-C, on-drop detection; (ii) EMC-Au, on-chip detection.

MB immunocomplexes were deposited onto the surface of the SPE-C and retained by a magnet, while the enzymatic substrate and electrochemical mediator (H_2_O_2_ and HQ) were added to perform the amperometric detection at −0.20 V. The detection potential was evaluated between zero and −0.3 V. The signal increased up to −0.2 V, keeping constant for larger negative potentials, as this one was the one we selected for the amperometric measurements.

For the EMC-Au approach, electrokinetic protocol for the injection of modified MB immunocomplexes and enzymatic substrate/electrochemical mediators as well as electrochemical detection were carefully studied as well (see Table 1).

The applied voltage, number of pulses and pulse time for the MB immunocomplex injection from the SR were assayed to place the optimum amount of MBs in the microchannel, in order to obtain the highest signal without clogging the channel. Electrokinetic conditions for the washing step were also studied to eliminate non-magnetically retained MBs, improving the assay precision. Then, the enzymatic substrates were continuously driven at +1500 V for 200 s. Under these optimal electrokinetic conditions, the detection potential was assayed between −0.10 V and −0.30 V. The highest signal/noise features were obtained at −0.20 V, which was chosen as the optimum detection potential.

### 3.2. Analytical Characteristics 

Analytical performance was carefully evaluated in both SPE-C and EMC-Au approaches. The calibration curves are depicted in Figure 3, while their corresponding analytical characteristics are summarized in Table 2.

From the obtained results, it is important to remark that both immunoassay detection approaches (SPE-C and EMC-Au) enabled PCT determination at the clinical significance levels needed for sepsis diagnosis and monitoring (LOD < cutoff). Interestingly, EMC-Au detection provides a lower LOD (calculated with a 3 S/m criteria where S is the standard deviation of the lowest assayed concentration (n = 10) and m is the slope of the calibration plot), while the SPE-C detection offers a wider working range (see Figure 3 and Table 2). This aspect can be attributed to the difference in the number of MBs trapped onto the SPE-C and the dimensions and material of the working electrode (C electrode, Ø = 4 mm) compared to that of the microfluidic chip (Au electrode, w = 100 µm). Moreover, a relevant comparative aspect deals with the shorter analysis time in the microfluidic chip, together with the possibility to automate the process, which enhances its potential as a point-of-care (POC) device. Indeed, taking into account that only a small fraction of the MBs from the sample reservoir are introduced into the main channel for detection each time, the analysis can be automatically repeated several times by programming the corresponding electro-kinetically driven protocol; (washing out the MBs from the previous run and placing a new batch into the central channel without any external intervention for a next run). The same MB immunocomplex batch (using 25 µL of sample) can be consecutively measured five times with good intra-assay precision (CV < 5%) in just 40 min. However, in the case of the SPE-C approach, to obtain five replicates, 125 µL of the sample and around 100 min would be needed.

PCT concentrations at two levels were used to evaluate the intra-assay (0.5 and 0.1 ng mL^−1^) and inter-assay precision (1000 and 20 ng mL^−1^) for on-drop SPE-C and on-chip EMC-Au, respectively (Table 2). In the case of SPE-C detection, the intra-assay and inter-assay precisions (n = 5) gave CV values below 8% in both cases. For EMC-Au, the intra-assay precision gave CV values of 5% (n = 5, same batch of MB immunocomplexes). The inter-assay precision for different MBs batches gave CV values of 9%.

The selectivity of the immunosensing configuration for PCT analysis was checked in the presence of a large excess of C-reactive protein (CRP) (16 µg mL^−1^, another biomarker usually determined for sepsis diagnosis), heparin (1 mg mL^−1^), ethylenedinitrilotetraacetic acid (EDTA) (1 mg mL^−1^) and citrate (0.15 M), as other relevant molecules that can coexist in blood samples. Without exception, cross-reactivity percentages lower than 1% were obtained. These results demonstrate the excellent selectivity of the PCT immunoassay. 

Due to its potential use as a POC, in order to simplify the entire procedure and, in turn, to reduce the final analysis times, the stability of the MB–captured antibody complexes was studied to be used as stock “reagents”. Their stability was studied at 4 °C during a period of 1 month using the on-drop SPE-C immunosensor approach. The control chart of the stability assay is shown in Figure 4, where each point corresponds to the mean value for three successive measurements performed in the same day (intra-day immunoassays). As can be seen, the immunosensor response remained inside the control limits placed at ±three times the standard deviation value calculated for the whole set of experiments, during the entire period of time checked (inter-day immunoassays, n = 14). These results demonstrate the excellent stability of the MB–cAb complexes.

### 3.3. Analysis of Human Serum and Plasma Samples

Analytical capabilities for PCT determination in clinical samples were also evaluated. Calibration curves were carried out in human serum and plasma from healthy individuals, using the on-drop SPE-C immunosensor. No matrix effect was observed after a comparison of the slope calibration plots obtained in PBS buffer with those obtained in both kinds of matrices. Indeed, identical slope calibration values (sensitivities) of 4300 ± 200, 4400 ± 400 and 4500 ± 200 nA ng^−1^ mL were obtained in buffer, serum and plasma, respectively. 

Then, accuracy was carefully studied using both detection approaches by recovery experiments conducted on both kinds of matrices spiked with relevant clinical levels of PCT. Table 3 demonstrates the suitability of the developed on-drop SPE-C and on-chip EMC-Au for PCT determination in human blood matrices at clinically relevant levels. It is important to remark that no single sample pretreatment was needed, due to the absence of matrix effects and the adequate working range of the immunoassay in both detection schemes. This aspect, which enhances the ease of use, together with the automation of the detection step and the portable characteristics of the devices, make the developed approaches suitable for potential POC tools for PCT determination and its use for sepsis diagnosis.

As previously mentioned in the introduction section, in recent years, significant effort has been focused on the development of new approaches for PCT determination. However, the potential of immunosensors based on magnetic beads has not been widely explored for PCT determination, which is apparent in the low number of publications (Table 4). In comparison with those works, our SPE-C approach offers enough sensitivity (similar to our previous work [40]) to perform reliable PCT detection, but with a significant reduction in the analysis time and sample volumes [16,17,35,36]. Moreover, the easy automation of the electrochemical transduction and the improved sensitivity [40] bring our new approach (on-chip EMC-Au) closer to the POC concept.

On the other hand, our on-chip EMC-Au approach has demonstrated to be a promising analytical strategy for PCT determination. It couples a magneto immunoassay and an electrochemical microfluidic chip under controlled electrokinetics. This approach presents advantages such as the easy miniaturization and integration of all system elements, fulfilling the POC requirements. As can be observed in Table 5, our approach is highly competitive in terms of sensitivity, analysis time and sample volume with previous on-chip non electrochemical-based approaches reported in the literature. 

## 4. Conclusions

A magnetic bead-based immunoassay using both on-drop SPE-C and on-chip EMC-Au electrochemical detection approaches have exhibited an excellent analytical performance for PCT determination, allowing for its determination in the clinically relevant concentration range, using very short analysis times and a low volume of serum and plasma samples. Both detection technologies are complementary. While SPE-C was simpler, the EMC-Au approach permits greater control and easier automation of the process, constituting an even more reliable alternative diagnostic tool for on-site/bed-site clinical analysis.

Both investigated approaches have demonstrated excellent biosensing capabilities for the simple and accurate determination of PCT in human samples when only small sample volumes are accessible. Therefore, these results reveal the analytical potential of highly miniaturized electrochemical devices in the field of PCT biosensing, one of the most important sepsis protein biomarkers.

## Figures and Tables

**Figure 1 biosensors-10-00066-f001:**
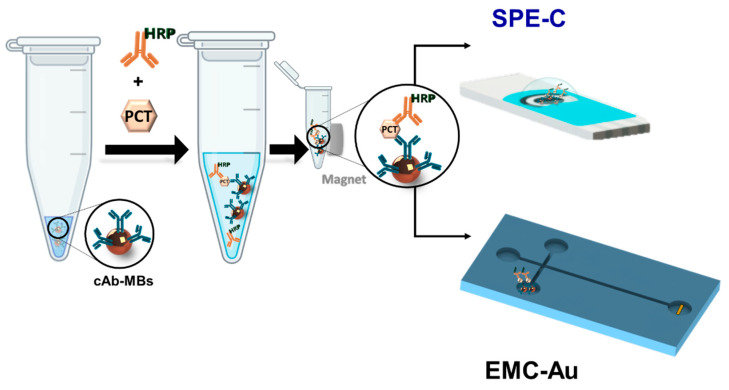
Magnetic bead-based electrochemical immunoassays—on-drop screen-printed carbon electrodes (SPE-C) and on-chip electro-kinetically driven microfluidic chips with integrated Au electrodes (EMC-Au).

**Figure 2 biosensors-10-00066-f002:**
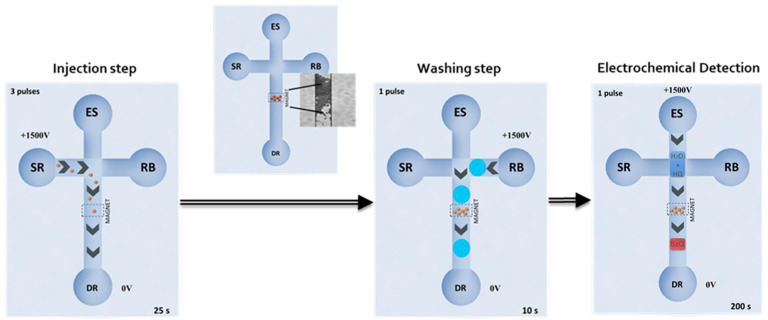
Electrokinetic protocol and electrochemical detection on EMC-Au. Sample reservoir (SR), running buffer (RB), enzymatic substrate reservoir (ER) and detection reservoirs (DR).

**Figure 3 biosensors-10-00066-f003:**
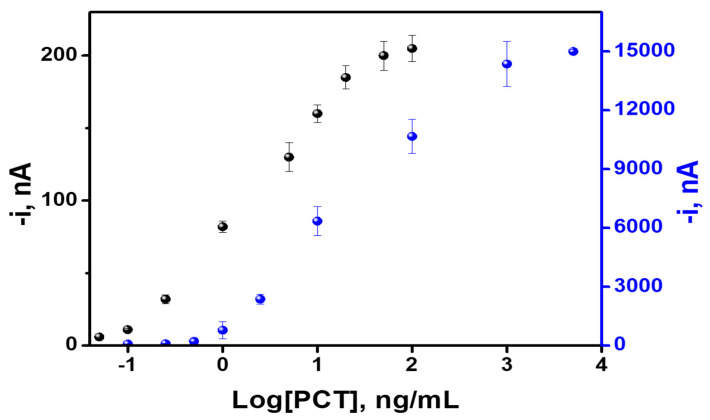
Calibration curves for PCT determination using both approaches: on-drop SPE-C and on-chip EMC-Au. The results are expressed as Mean values ± standard deviation (n = 3). The error bar corresponds with the standard deviation.

**Figure 4 biosensors-10-00066-f004:**
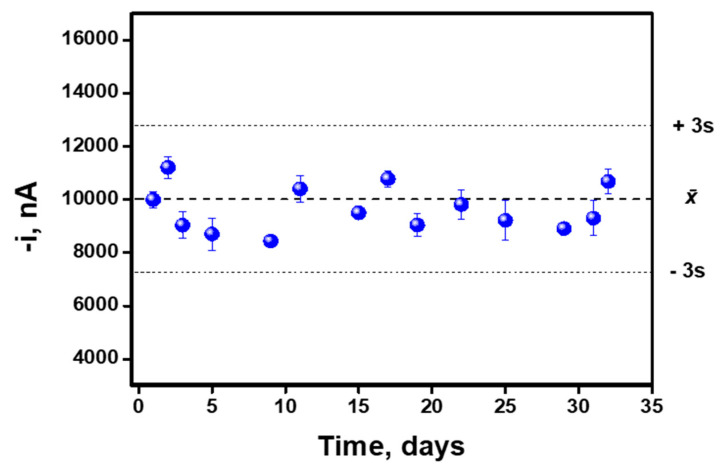
Stability of the MB–cAb complexes. Central and limit lines correspond to average ± three times the standard deviation (n = 14) obtained for inter-day immunoassays. Each individual point corresponds to the average ± standard deviation (n = 3) obtained for the intra-day immunoassays.

**Table 1 biosensors-10-00066-t001:** Electrokinetics and electrochemical detection, EMC-Au optimizations.

Step	Parameter	Studied Range	Selected Value
Immunocomplex-MB Injection	Immunocomplex-MB dilution (*v*/*v*)	1:10–1:200	1:10
	Applied Voltage (V)	+1000–2000	+1500
	Number of pulses	1–5	3
	Pulse time (s)	10–50	25
Washing	Applied Voltage (V)	+1000–2000	1500
	Number of pulses	1–5	1
	Pulse time (s)	10–50	25
Enzyme substrates pumping/driven	Applied Voltage (V)	+1000–2000	+1500
	Time (s)	----	200 s
Detection	E (V)	−0.10–(−0.30)	−0.20

**Table 2 biosensors-10-00066-t002:** Analytical characteristics for procalcitonin (PCT) determination using both approaches: SPE-C and EMC-Au.

Analytical Characteristic	SPE-C	EMC-Au
EC_50_, ng mL^−1^	20.2	2.2
Working range, ng mL^−1^	0.5–1000	0.1–20
r	0.990	0.990
LOD, ng mL^−1^	0.1	0.04
Intra-assay, CV%	<7.5%	5%
Inter-assay, CV%	8%	9%

**Table 3 biosensors-10-00066-t003:** PCT determination in human plasma and serum samples.

PCT_added_ (ng/mL)	SPE-C				EMC-Au	
	Serum		Plasma		Serum	
	PCT_found_ (ng/mL)	Recovery (%)	PCT_found_ (ng/mL)	Recovery (%)	PCT_found_ (ng/mL)	Recovery (%)
Before spiked	<0.1	---	<0.1	---	<0.04	---
1.0	0.9	90 ± 3	1.2	120 ± 6	1.1	110 ± 5
10.0	8.9	89 ± 14	10.4	104 ± 7	9.6	96 ± 2
100.0	99	99 ± 7	101	101 ± 2	--- *	---

* Sample was not directly measured, since its concentration is beyond the working range.

**Table 4 biosensors-10-00066-t004:** Overview of magneto-immunosensors for PCT determination.

Technique	cAb Immobilization	Assay Format	dAb Label	WR	LOD	Analysis Time *	Sample Volume	Sample	Ref.
**Chemiluminescence**	MB–(anti-FITC–Ab)/FITC–cAb	Double Sandwich	dAb–ABEI	0.09–600 ng mL^−1^	30 pg mL^−1^	25 min	40 µL	Serum	16
**Chemiluminescence**	MB–COOH/cAb	Sandwich	dAb–(PS-ALP)	1–10^4^ pg mL^−1^	0.045 pg mL^−1^	1 h	800 µL	Serum	17
**UV-vis Spectroscopy**	MB–COOH/cAb	Sandwich	dAb–HRP	0.1–10 ng mL^−1^	40 pg mL^−1^	1.5 h	100 µL	Serum	35
**UV-vis Spectroscopy**	MB–COOH/cAb	Sandwich	dAb–(AuNPs-HRP)	0.02–20 ng mL^−1^	20 pg mL^−1^	1.5 h	50 µL	Serum	36
**Amperometry**	MB–Streptavidin/Biotin–cAb	Sandwich	dAb–HRP	0.25–100 ng mL^−1^	50 pg mL^−1^	20 min	25 µL	Neonates plasma	40
**Amperometry**	MB–Streptavidin/Biotin–cAb	Sandwich	dAb–HRP	0.5–1000 ng mL^−1^	100 pg mL^−1^	20 min	25 µL	Serum Plasma	Our work (SPE-C)

* Analysis time is measured after captured Ab immobilization stage. Abbreviations used: antibody (Ab); captured antibody (cAb); detection antibody (dAb); working range (WR); limit of detection (LOD); magnetic beads (MBs); horseradish peroxidase (HRP); fluorescein isothiocyanate (FITC); (aminobutyl)-N-(ethylisoluminol) (ABEI-N); alkaline phosphatase (ALP); polystyrene microsphere (PS); gold nanoparticles (AuNPs).

**Table 5 biosensors-10-00066-t005:** Overview of microfluidics immunoassays for PCT determination.

Technique	cAb Immobilization	Assay Format	dAb Label	WR	LOD	Analysis Time *	Sample Volume	Sample	Ref.
**Fluorescence**	Covalent cAb	Sandwich	dAb–DY647	0.7–25 ng mL^−1^	0.2 ng mL^−1^	23 min	100 µL	Serum	26
**Reflection Fluorescence**	Covalent cAb	Sandwich	dAb–DY647	5–500 ng mL^−1^	1 ng mL^−1^	11 min	10–75 µL	Serum Plasma	28
**Fluorescence**	Adsorption cAb	Sandwich	dAb–DY647	640–3400 ng mL^−1^	50 ng mL^−1^	22 min	280 µL	Serum Diluted 1:10	29
**Reflection Fluorescence**	Adsorption cAb	Sandwich	dAb–cyanine	0.06–7.18 ng mL^−1^	0.02 ng mL^−1^	<9 min	50 µL	Serum Plasma Whole blood	30
**Nanoplasmonic**	Covalent cAb	Sandwich	dAb–AuNPs	1 pg mL^−1^–100 ng mL^−1^	95 fg mL^−1^	<15 min	---	Serum	39
**Amperometry**	MB–Streptavidin/Biotin–cAb	Sandwich	dAb–HRP	0.1–20 ng mL^−1^	40 pg mL^−1^	20 min	25 µL	Serum Plasma	Our work (EMC-Au)

* Analysis time is measured after captured Ab immobilization stage. Abbreviations used: antibody (Ab); captured antibody (cAb); detection antibody (dAb); working range (WR); limit of detection (LOD); magnetic beads (MBs); horseradish peroxidase (HRP); fluorescein isothiocyanate (FITC); (aminobutyl)-N-(ethylisoluminol) (ABEI-N); alkaline phosphatase (ALP); polystyrene microsphere (PS); gold nanoparticles (AuNPs).

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
