# Peer review of "Magnetic Bead-Based Electrochemical Immunoassays On-Drop and On-Chip for Procalcitonin Determination: Disposable Tools for Clinical Sepsis Diagnosis"

_biosensors, 2020, doi:10.3390/bios10060066_

Round 1

Reviewer 1 Report

 This paper reports a method of electrochemical measurement for Procalcitonin (PCT) using antigen-antibody reaction supported by Magnetic beads and disposal screen printed electrode or microchip. The method is standard and not so novel. However, authors successfully establish a series of procedure to measure PCT in serum and plasma with sufficient accuracy and measurable range with low-cost, quick and capable for point-of-care testing, with comparison of previous works described in this paper. I think this is worth to publish after following modification.

  1. The detail of electrochemical measurement should be provided. Especially, which type of amperometry did they use? And how the current value is calculated from the row data? The typical row data for PCT measurement with blank sample should be indicated in supplemental file. This is helpful to appeal the quality of measurement.

  1. Please make clear the method to calculate the error bar in figure 3.

  1. Add the definition of HQ.

  1. Use same expression, mg/mL or mg mL-1, DropSens or Dropsens.

Author Response

Reviewer overall comment:

This paper reports a method of electrochemical measurement for Procalcitonin (PCT) using antigen-antibody reaction supported by Magnetic beads and disposal screen printed electrode or microchip. The method is standard and not so novel. However, authors successfully establish a series of procedure to measure PCT in serum and plasma with sufficient accuracy and measurable range with low-cost, quick and capable for point-of-care testing, with comparison of previous works described in this paper. I think this is worth to publish after following modification.

Author´s response:

We are glad to read the comment from the reviewer. Please, note that the main novelty of this work has been to explore the analytical potential of (highly) miniaturized electrochemical devices in the field of PCT bio sensing, one of the most important sepsis protein biomarker. While immunoassay is standard, electrochemical detection remains to be studied for this important sepsis biomarker. In this sense, the EMC-Au approach is explored here by the first time.

The authors have considered the comments and suggestions for strengthening this new submitted manuscript.

Reviewer comment:

  1. The detail of electrochemical measurement should be provided. Especially, which type of amperometry did they use? And how the current value is calculated from the row data? The typical row data for PCT measurement with blank sample should be indicated in supplemental file. This is helpful to appeal the quality of measurement.

Author´s response and action:

Thank you for your comment.

The amperometric signals obtained at -0.20 V were calculated as the difference between the steady state and the background currents at 200s. These signals were fitted to the four-parameter logistic equation using the software SigmaPlot 10.

As referee suggested, a new Figure S2 has been included in supplementary information including the row data for the signals provided by the blank sample and a measurable PCT concentration.

Reviewer comment:

  1. Please make clear the method to calculate the error bar in figure 3.

Author´s response and action:

Thank you for your comment.

The results are expressed as Mean values ± standard deviation (n=3). The error bar corresponds with the standard deviation. It has been now included in the caption of Figure 3 in new manuscript.

Reviewer comment:

  1. Add the definition of HQ.

Author´s response and action:

Thank you for your comment.

This point has been included in the section 2.1 (line 82) of the new version of the manuscript.

Reviewer comment:

  1. Use same expression, mg/mL or mg mL-1, DropSens or Dropsens.

Author´s response and action:

Thank you for your comment. We are sorry for these typos.

The new manuscript has been carefully revised.

Reviewer 2 Report

The manuscript biosensors-801559 by Águeda Molinero-Fernández et al. reports on the detection of procalcitonin (PCT) by two approaches of electrochemical biosensing. The topic is of high interest; unfortunately, the manuscript currently fails to convince the reader about the novelty and advantages of the proposed approach. However, when this point, together with other major revisions (see below), is successfully addressed, the manuscript can be reconsidered for publication.

1. (i) The manuscript builds on the previous work of the authors (e.g., DOI 10.1021/acssensors.9b00890); however, this is not sufficiently pointed out neither in the Introduction nor in the Discussion. The LOD, as well as the analysis time, is very similar; does the slightly improved procedure in the current manuscript bring any significant advantages? (ii) Similarly, the manuscript has to discuss the advantages and disadvantages of the proposed method with previous work published by other authors.

2. The Results and discussion part is showing the calibration curves after the optimization but completely omits the amperometric curves and results from the optimization experiments. This data has to be included either in the manuscript or in the Supporting information.

3. (i) The details on the LOD calculation have to be provided. (ii) Calibration plots in Figure 3 should be fitted, e.g., with logistic function.

4. In case of Figure 4, the lines corresponding to three times the standard deviation appear significantly further than three times the typical error bars show in the graph. The authors should explain this.

5. (i) What method was used to prove that the real samples had “undetectable PCT concentration”? (ii) The recovery calculations (Table 3) should not only compare the achieved results with the known spiked amount, but also with the independent reference method. (iii) Furthermore, the analysis of the non-spiked sample should be included. (iv) In the case of 10 ng/mL of spiked PTC analyzed using SPE-C, the found concentration of 8.9 ng/mL currently corresponds to the recovery of 88% (Table 3). Is this correct?

6. The COVID-19 related paragraph in the Introduction seems out of context. Either this has to be removed, or the rest of the manuscript has to be adapted to discuss the performance of the sensors in connection to sepsis connected with COVID-19.

7. Other minor editing work is necessary. (i) “KDa” on Page 2, Line 46 should be written with lowercase “k”. (ii) The correct name of PBS (Page 3, Line 88) is phosphate-buffered saline. (iii) Either “min” or “minutes” should be used consistently across the whole manuscript. (iv) Exact value instead of “<8%” should be provided in Table 2 for Intra-assay variability. (v) The Conclusions should be prolonged to provide more information. (vi) The information on particular author contributions is missing. (vii) The journal names formats are currently randomized between full names, abbreviations with full stops, and abbreviations without full stops. This should be unified. (viii) The numbers in chemical formulas in references 52 a 55 are not written appropriately as lower indexes.

Author Response

Reviewer overall comment:

The manuscript biosensors-801559 by Águeda Molinero-Fernández et al. reports on the detection of procalcitonin (PCT) by two approaches of electrochemical biosensing. The topic is of high interest; unfortunately, the manuscript currently fails to convince the reader about the novelty and advantages of the proposed approach. However, when this point, together with other major revisions (see below), is successfully addressed, the manuscript can be reconsidered for publication.

Author´s response:

Thank you for your comment related to high interest of the topic we have work in. Please, note that the main novelty of this work has been to explore the analytical potential of (highly) miniaturized electrochemical devices in the field of PCT bio sensing, one of the most important sepsis protein biomarker. While immunoassay is standard, electrochemical detection remains to be studied for this important sepsis biomarker. In this sense, the EMC-Au approach is explored here by the first time.

The authors have considered the comments and suggestions for strengthening this new submitted manuscript.

Related to the novelty and advantages of our approach in comparison to previously published works (depicted in Table 4 and Table 5) to determine PCT, we would like to point out some considerations. First of all, our work is the only one dealing with electrochemical detection of PCT. It is known that electrochemistry is very easy to miniaturize in comparison with optical detection without loss in its performance. This aspect together with the possibility of automation of the on-chip EMC-Au approach, enhance its potential as a POC device. No one of the published works is based on an electrokinetically driven microfluidic chip, which stands out by the automated and strict control of fluids and analytical stages, just using a few platinum wires and power source.  Besides, the shorter analysis time and, specially, the low volume of sample used (the lowest of the published works) improve clearly its merits. All of these features are added to reliable measurement in the clinically relevant concentrations range of PCT. In this sense, we think that the developed work is clearly competitive with the existent published articles.

Reviewer comment:

  1. (i) The manuscript builds on the previous work of the authors (e.g., DOI 10.1021/acssensors.9b00890); however, this is not sufficiently pointed out neither in the Introduction nor in the Discussion. The LOD, as well as the analysis time, is very similar; does the slightly improved procedure in the current manuscript bring any significant advantages? (ii) Similarly, the manuscript has to discuss the advantages and disadvantages of the proposed method with previous work published by other authors.

Author´s response and action:

Thank you for your comment.

Comparison with other magneto-immunosensors for PCT determination, including our previously dual PCT/CRP immunosensor, is depicted in table 4. It is important to take in mind that our previously developed immunosensor allows the simultaneous determination of two key biomarkers in sepsis diagnosis (PCT and C-reactive protein -CRP-). Since both works (ACS Sensors 2019 and this work) include determination of PCT, the used reagents are the same. However, specific optimization experiments were developed for each work, mainly for detection, since electrode geometries (dimensions and shapes) and conditions are different. The aim of both works is also different. As previously stated, ACS sensors 2019 was focused on the simultaneous determination of two key sepsis biomarkers and their determination in very special samples as those coming from neonates with sepsis suspicion. In the work reported here, the primary objective was devoted to the comparison of two different electrochemical detection approaches (on-drop SPE-C and on-chip EMC-Au) to develop a reliable alternative diagnostic tool for on-site clinical analysis. In this sense, although both detection approaches enabled the PCT determination at the clinical significance levels needed for sepsis diagnosis and monitoring (LOD < Cut-off), they also present remarkable differences. Besides the LOD and working range differences, it is important to remark the shorter analysis time in the on-chip EMC-Au, together with the possibility to automate the process, enhancing its potential as POC device. Indeed, taking into account that only a small fraction of the MBs from the sample reservoir are introduced into the main channel for detection each time, the analysis can be automatically repeated several times by programming the corresponding electro-kinetically driven protocol; (washing out the MBs from the previous run and placing a new batch into the central channel without any external intervention for a next run). The same MBs-immunocomplex batch (using 25 µL of sample) can be measured 5 times with good intra-assay precision (CV < 5%) in just 40 minutes. However, in the case of the SPE-C approach, to obtain 5 replicates, 125 µL of sample and around 100 minutes would be needed.

Related to previously reported magneto-immunosensors for PCT determination by other authors, any of them is based on electrochemical detection. As can be observed in Table 4 and the text (lines 261-268) these works present significant higher analysis times and sample volumes in comparison with the present work. Although one of them (ref. 23) presents remarkable better LOD, our work offers enough sensitivity and concentration working range for the clinicians to perform a reliable sepsis diagnosis.

We think that the authors provided in the original version two well-separated detailed tables (4 and 5), offering an accurate comparison of our results with relevant literatures, for both SPE-C and MCE-Au approaches. Our manuscript contains 64 references. Anyway, following referee suggestion, the discussion has been improved.

In addition, we would like to underline that the chip approach is novel and full electrokinetics optimization has been addressed. This aspect seems overlooked by the reviewer.

Reviewer comment:

  1. The Results and discussion part is showing the calibration curves after the optimization but completely omits the amperometric curves and results from the optimization experiments. This data has to be included either in the manuscript or in the Supporting information.

Author´s response and action:

Thank you so much for your comment.

A new Figure S2 has been included in supplementary information including the row data for the signals provided by the blank sample and a measurable PCT concentration as reviewer 1 required too.

Optimization of the antibodies concentration (capture and detection antibodies) and incubation times has also been now added as Figures S3 and S4, respectively.

Reviewer comment:

  1. (i) The details on the LOD calculation have to be provided. (ii) Calibration plots in Figure 3 should be fitted, e.g., with logistic function.

Author´s response and action:

Thank you for your comment.

The amperometric signals at -0.20 V were calculated as the difference between the steady state and the background currents at 200 s, and fitted to the four-parameter logistic equation using the software SigmaPlot 10.0. (Equation 1)

Equation 1 (See the attachment)

Where imax and imin are the maximum and minimum current values of the calibration graph; EC50 value is the analyte concentration corresponding to a 50% of the maximum signal; and h is the hill slope.

The LOD was calculated as 3 S/m where S was the standard deviation of the lowest assayed concentration (n=10) and m is the slope of the calibration plot.

These issues have been added in the new manuscript and the EC50 value has also been added in table 2.

Reviewer comment:

  1. In case of Figure 4, the lines corresponding to three times the standard deviation appear significantly further than three times the typical error bars show in the graph. The authors should explain this.

Author´s response and action:

Thank you for your comment.

In the control chart of the stability assay (shown in Figure 4), while the average and limit lines (± 3s) are calculated from inter-day immunoassays (n=14), the individual data are from intra-day immunoassays (n=3).

This point has been now clarified in the revised manuscript. A revised paragraph discussing Figure 4 and revised caption for Figure 4 is included.

“Due to its potential use as a POC, in order to simplify the entire procedure and in turn to reduce the final analysis times, the stability of the MBs-capture antibody complexes was studied to be used as stock "reagents". Their stability was studied at 4 °C during a period of 1 month using the on-drop SPE-C immunosensor approach. The control chart of the stability assay is shown in Figure 4, where each point corresponds to the mean value for three successive measurements performed in the same day (intra-day immunoassays). As it can be seen, the immunosensor response remained inside the control limits placed at ±3 x the standard deviation value calculated for the whole set of experiments, during the entire period of time checked (inter-day immunoassays, n=14). The results demonstrated an excellent stability of the modified MBs-capture antibody complexes”.

Reviewer comment:

  1. (i) What method was used to prove that the real samples had “undetectable PCT concentration”? (ii) The recovery calculations (Table 3) should not only compare the achieved results with the known spiked amount, but also with the independent reference method. (iii) Furthermore, the analysis of the non-spiked sample should be included. (iv) In the case of 10 ng/mL of spiked PTC analyzed using SPE-C, the found concentration of 8.9 ng/mL currently corresponds to the recovery of 88% (Table 3). Is this correct?

Author´s response and action:

Thank you so much for your comments.

(i) Healthy volunteer samples were analyzed by our two methods to check the absence ([PCT] < LODs) of PCT.

(ii) Due to reference material for PCT is not available; the recovery is a well-established approach to evaluate the accuracy.

It is true that the use of a free interference method is pertinent to evaluate the accuracy; under these circumstances since not CRM for PCT is available. However, in our case, it is not necessary. Indeed, there are excellent agreements between the results obtained by both proposed methods (on drop, on chip). Therefore, from our humble point of view, it is well enough to assess the accuracy of the proposed approaches by recovery assays. It is also in agreement with the referee 1´s opinion who highlighted that “….authors successfully establish a series of procedure to measure PCT in serum and plasma with sufficient accuracy…”

(iii) The authors proved the absence of PCT because the clinical sample was measured with both approaches and the signal was non discernible from PBST. Also, these results were in agreement with those obtained in the recovery experiments.

(iv)The recovery for 8.9 ng/mL was of 89 %, sorry for this mistake.

Reviewer comment:

  1. The COVID-19 related paragraph in the Introduction seems out of context. Either this has to be removed, or the rest of the manuscript has to be adapted to discuss the performance of the sensors in connection to sepsis connected with COVID-19.

Author´s response and action:

Thank you so much for your comment.

COVID-19 related paragraph has been removed in the revised .manuscript

Reviewer comment:

  1. Other minor editing work is necessary. (i) “KDa” on Page 2, Line 46 should be written with lowercase “k”. (ii) The correct name of PBS (Page 3, Line 88) is phosphate-buffered saline. (iii) Either “min” or “minutes” should be used consistently across the whole manuscript. (iv) Exact value instead of “<8%” should be provided in Table 2 for Intra-assay variability. (v) The Conclusions should be prolonged to provide more information. (vi) The information on particular author contributions is missing. (vii) The journal names formats are currently randomized between full names, abbreviations with full stops, and abbreviations without full stops. This should be unified. (viii) The numbers in chemical formulas in references 52 a 55 are not written appropriately as lower indexes.

Author´s response and action:

Thank you for your comment. Conclusion section has been revised in order to provide more information. We really appreciate this important comment.

We are sorry for the typos. The new version of the manuscript has been carefully revised.

Round 2

Reviewer 2 Report

The authors have successfully addressed my comments. The manuscript can be recommended for acceptance in the present form.